# Willingness and ability to pay for breast cancer treatment among patients from Addis Ababa, Ethiopia: A cross-sectional study

Tamiru Demeke[1,2,3], Damen Hailemariam[2], Pablo Santos[1,3], Edom Seife[4], Adamu Addissie[2], Eric Sven Kroeber[1,3,5], Rafael Mikolajczyk[1], Birgit Silbersack[5], Eva Johanna Kantelhardt[1,3,6‡], Susanne Unverzagt[3,5‡]*

1 Institute of Medical Epidemiology, Biostatistics and Informatics, Martin Luther University Halle-Wittenberg, Halle (Saale), Germany, 2 School of Public Health, College of Health Sciences, Addis Ababa University, Addis Ababa, Ethiopia, 3 Global and Planetary Health Working Group, Martin Luther University Halle-Wittenberg, Halle (Saale), Germany, 4 Radiotherapy Centre, Addis Ababa University, Addis Ababa, Ethiopia, 5 Institute of General Practice and Family Medicine, Martin Luther University Halle-Wittenberg, Halle (Saale), Germany, 6 Department of Gynecology, Martin Luther University Halle-Wittenberg, Halle, Germany

‡ EJK and SU are shared senior authorship
* susanne.unverzagt@uk-halle.de

**Data Availability Statement:** All relevant data are within the paper and its Supporting Information files. The minimal dataset dataset data required to

## Abstract

### Introduction

Breast cancer (BC) is the most common malignant neoplasm among women in Addis Ababa, Ethiopia. The willingness and ability to pay (WATP) for treatment is a contributing factor in the utilization of health care services. The World Health Organization Breast Cancer Initiative calls for 80% of patients to complete multimodality treatment and indicates payment as central factor to improve BC outcome. The objectives of this study are to identify how much female BC patients paid in Addis Ababa for BC treatment, their WATP for BC treatment, and the factors that affect WATP.

### Methods

The researchers collected data from 204 randomly selected BC patients who were treated in one of four different health facilities (one public and three private) between September 2018 and May 2019. A structured questionnaire was used to assess their WATP for BC treatment and multivariable regression to investigate factors associated with patients' WATP.

### Results

Of interviewed patients, 146 (72%) were at reproductive age. Patients' median expenditure for all BC treatment services was 336 US dollars (USD) in a public cancer center and 926 USD in privately owned health facilities. These amounts are in contrast with a reported WATP of 50 USD and 149 USD. WATP increased with increasing expenditure (OR 1.43; 95% CI 1.09 to 1.89 per 100 US), educational level (OR 1.37; 95% CI 1.02 to 1.85) and

replicate all study findings reported in the article can be found under doi: 10.6084/m9.figshare.25370860.v2 The python script showing all steps to reproduce data cleaning, model building, regression coefficients with 95%CI and and p-values can be found under doi: 10.6084/m9.figshare.25370950.v1.

**Funding:** This work was supported by DAAD PAGEL project No. 57513614. It was also supported by Else-Kroener-Foundation through Martin-Luther-University, Halle-Wittenberg, Germany, Grant No. 2018_HA31SP. The work was also supported through the German Ministry of Research and Education, Grant 01KA2220B. The funders had no role in study design, data collection and analysis, decision to publish, or preparation of the manuscript.

**Competing interests:** The authors have declared that no competing interests exist.

service quality (OR 1.34; 95% CI 1.04 to 1.72). In contrast, a monthly income increase by 100 USD corresponds to a 17% decrease of WATP (OR 0.83; 95% CI 0.70 to 0.99).

## Conclusions

We demonstrated that BC treatment was very expensive for patients, and the cost was much higher than their WATP. Thus, we suggest that BC should be included in both social and community-based health insurance plans and treatment fees should consider patients' WATP.

## Introduction

Breast cancer (BC) is a growing public health challenge in Ethiopia [1,2]. According to data from the Addis Ababa Population Based Cancer Registry, the cumulative BC incidence in the capital city alone reached 4,500 cases as of 2019 [3]. The GLOBOCAN 2020 report indicated that in Ethiopia there were 16133 new BC cases and 9061 deaths due to BC. These figures put the country in the third rank among African countries [4].

Many breast cancer patients seek treatment at the Tikur Anbessa Specialized Hospital-Radiation Centre (hereafter called Public Cancer Centre [PCC]) to get treatment as, until 2022, it was the first and only government owned hospital that provided cancer treatment including radiotherapy. In addition, there are three privately owned health facilities in Addis Ababa that provide oncology service but are limited to chemotherapy only [5]. Among patients who presented in PCC for treatment, almost 71% were diagnosed at advanced stages due to lengthy intervals between the time of recognition of symptoms and getting treatment [6]. Apart from various medical and non-medical factors, many BC patients presented in both PCC and private health facilities (PHFs) at advanced stages due to low income and the high costs of BC treatment. Among these patients, some discontinued their treatment, frequently due to shortage of money [7–9].

Ethiopia's national health accounts reports indicated that out-of-pocket (OOP) expenditure accounts for about 30% of the total health expenditure [10]. This is also applicable to BC patients. The heavy reliance on OOP expenditure triggered policy makers to pay attention to the cost of health care services because a large OOP expenditure can make the health expenditure catastrophic [11]. An OOP expenditure is said to be catastrophic if it is above the estimated threshold share of household expenditure, which can result in patients being forced to sacrifice other basic needs, sell assets, incur debt, or be impoverished [12,13] or if the OOP expenditure is equal to or greater than 40% of non-food expenditure or capacity to pay (CTP). CTP is the difference between subsistence expenditure and monthly household expenditure (i.e., consumption [14]. This in turn will compel patients and their families to reduce consumption of basic necessities and this further will push them into poverty. Thus, the Federal Democratic Republic of Ethiopia (FDRE) Ministry of Health (MOH) prepared and implemented a five-year national cancer control program to reduce the financial burden [15].

Affordability of BC treatment is an important factor for optimal treatment compliance. The WHO Global Breast Cancer Initiative has recommended that 80% of patients should complete multimodal treatment with an affordable charge as an essential component for success [16]. To determine the affordability of treatment, it is paramount to know what patients are willing and able to pay (WATP) for treatment. Willingness to pay (WTP) refers to a person's willingness to pay for desired goods or services, while ability to pay (ATP) refers to their financial

capacity to do so. [17]. Therefore, it is recommended to combine these two concepts and use WATP to estimate the real demand for goods and services including health care services [18–22]. Accordingly, many studies assessed the demand for BC treatment using either the contingent valuation (CV, see Methods) technique for stated preferences or the conjoint analysis (CA) technique for revealed preference. The CV technique is generally recommended for health economics research [19,23–28]. The method has been validated and employed in several similar studies [20,29,30].

It is a survey approach designed to fill the gap in the market for public goods by asking how much money people are willing to pay (WTP) for specified goods or services including health care services [31].

Analyzing WATP is crucial for understanding the demand for health care services and provides evidence to both policy makers and health professionals regarding the determination of health care service fees [32], to guiding financial strategies to ensure access to recommended BC treatments for all patients, and providing treatment based on patients' economic [33–35].

The aims of this study were to (i) identify how much female patients were paying for their treatment both in public and private health facilities, (ii) investigate the WATP of BC patients living in Addis Ababa, and (iii) identify the factors that influenced their WATP.

## Materials and methods

### Study population

We conducted a quantitative cross-sectional study for which data was collected from BC patients who presented in one of four selected health facilities (one public and three privately owned) in Addis Ababa, Ethiopia. Data collection was done between September 2018 and May 2019. Data analysis was performed between April 2020 and January 2021. For this study, For this study, we included all BC patients permanently residing in Addis Ababa and receiving treatment at PCC and the three privately owned health facilities at the time of data collection. The sample design was simple random sampling that is BC patients who came to the aforementioned health facilities were randomly selected among those BC patients who came to the health facilities at the time of data collection.

Sample size calculation was done based on accuracy (two-sided width of the 95% confidence interval CI) of the estimate and an assumed standard deviation of direct medical costs of 355 USD per BC treatment, requiring a distance from the mean to both directions of 50 (width of the CI of 100) [36], resulting in a sample size of n = 194. In addition, 15% was added for non-response rate and the sample size became 223 patients. This sample size was distributed among the four health facilities in proportion to the number of BC patients they treated before data collection time using simple proportional allocation method. Accordingly, 133 patients from PCC and 90 patients from the three private health facilities were contacted.

### Data collection

The data collection tool was a structured questionnaire consisting of the following sections (WHO): socio-economic, (1) history of BC and current health status (2), expenditures related to BC treatment, and (AAPCR) patients' WATP for BC treatment. To measure WATP, patients were asked how much they would be willing and able to pay for each health care service they receive based on their present experience, without considering the frequency of the service or the number of chemotherapy cycles. The questionnaire was prepared based on the CV technique that consists of open-ended and close-ended questions to ask patients the maximum amount they were WATP for their treatment.

## Statistics

The data were processed with SPSS version 26, STATA version 15 and the Statsmodels library version 0.14.0 in Python [37]. Results of descriptive statistics were presented as absolute and relative frequencies, with the corresponding medians. We assessed the effects of sociodemographic and medical factors on patients' WATP while adjusting for confounders by using a multivariate regression model. Since WATP values did not follow a normal distribution, we avoided heteroscedasticity by fitting data as a generalized linear model (Gamma family with log link [38]). The exponentiated coefficients resulting from the multivariate regression were interpreted as Odds Ratios (ORs) [39,40]. The number of samples considered for the regression was the sample subset (n = 121) for which valid values were available for all assessed variables. Due to the lack of collinearity among the explanatory variables (Pearson's $\rho < 0.5$ for all variable pairs), imputations were not performed. The number of missing data points that led to the final sample subset of 121 patients assessed in the regression are given for each variable in Table 1.

## Ethics approval

The Institutional Review Board (IRB) of the College of Health Sciences at Addis Ababa University in Ethiopia provided ethical approval prior to the collection of study data under Protocol Number 051/18/SPH. Informed consent was obtained orally from each patient at the beginning of the interview and documented on the questionnaire by the data collectors.

## Results

### Descriptive statistics

Of a total of 223 patients who were asked for interview, 204 (91%) agreed to participate in the study. Of the 204 interviewed patients, 151 (74%) were treated at PCC and 53 (26%) at the three PHFs. Patients were between 23 and 75 years old and 72% of patients were within reproductive age. Half of the patients were unemployed or housewives, 11% of patients were unable to read and write, and 69% were married. The median family size excluding the interviewed patient was four (range 0–12). 72% of patients were treated for up to two years. The most common symptoms leading to a diagnosis with BC were breast swelling (61%) followed by breast pain (28%) (Table 1).

Of the interviewed patients, 72% were willing to disclose their average monthly families' income. Of these, 31% had a monthly income below 75 USD. There was a broad range of monthly income between 5 and 1,011 USD (Table 1). Participants in the study were grouped according to their monthly income based on the World Bank's Poverty Assessment of Ethiopia and Ethiopia's low income tax rate. Accordingly, an income of less than 1.25 USD per day was considered the poverty line. [41]. Monthly income greater than 38 USD but less than or equal to 75 USD is considered lower-middle income, monthly income greater than 75 USD but less than or equal to 150 USD is considered middle income, and monthly income greater than 150 USD is considered upper income.

Patients' income sources were, their own (26%), their husband's salary (23%), a combination of their own income and their husband's salary (21%), pension (16%) or support from their children or other relatives (14%). Regarding average monthly household expenses, excluding expenditure for BC treatment, 56% spent more than 75 USD, with a median expenditure of 98 USD.

### Expenditure and satisfaction with treatment

Most patients (82%) were satisfied with the treatment they received, while 57% of patients said the quality of the services was good or very good. However, 55% of patients deemed the treatments expensive.

**Table 1. Demographic characteristics of patients (n = 204).**

| Characteristic | n (%) * |
|---|---|
| Age (years) | |
| 23–49 | 146 (71) |
| 50–75 | 58 (28) |
| Employment status | |
| Unemployed or housewives | 101 (50) |
| Employed | 73 (36) |
| Self-employed | 26 (13) |
| Other | 4 (2) |
| Educational level | |
| Without elementary level | 54 (26) |
| Elementary or secondary school level | 70 (34) |
| Higher education | 78 (38) |
| Unknown | 2 (1) |
| Marital status | |
| Single | 23 (11) |
| Married | 138 (68) |
| Ex-married (widowed, separated, or divorced) | 43 (21) |
| Family size | |
| $\leq 3$ | 82 (40) |
| 4 to 7 | 100 (49) |
| $\geq 8$ | 19 (9) |
| Unknown | 3 (1) |
| Number of years with BC | |
| < 1 | 30 (15) |
| 1 | 67 (33) |
| 2 | 49 (24) |
| 3 | 46 (23) |
| No response | 12 (6) |
| Reasons for diagnosis | |
| A physician advised me to consult Oncologist about my BC when I went for other treatment | 15 (7) |
| Breast swelling | 118 (58) |
| Breast pain | 59 (29) |
| No response | 12 (6) |
| Average monthly income (USD) | |
| $\leq 38$ | 34 (17) |
| > 38 to $\leq 75$ | 29 (14) |
| > 75 to $\leq 150$ | 41 (20) |
| $\geq 150$ | 42 (21) |
| No response | 58 (28) |

BC = Breast cancer; USD = United States Dollar

* Sums may not add up to 100% due to rounding.

Patients treated at PCC paid a median of 336 USD (interquartile range [IQR] 97 to 711), whereas patients treated in PHFs paid a median of 926 USD (IQR 206 to 1,581) for all services. Among the services provided at PCC, patients paid the highest median amount of 152 USD (IQR 76 to 253) for chemotherapy, followed by 126 USD (IQR 48 to 253) for surgery, and 106 USD (IQR 29 to 253) for medication. Patients treated at PHFs paid the highest median amount

**Table 2. Comparison between PCC and PHFs concerning the amounts actually paid for each component of the BC treatment.**

| Service | Median (1) | | IQR (USD) | |
|---|---|---|---|---|
| | PCC | PHF | PCC | PHF |
| Consultation | 0.1 (112) | 10.1 (52) | 0.1–0.1 | 8.8–10.1 |
| Laboratory | 12.6 (90) | 35.4 (42) | 5.1–38.9 | 21.5–97.9 |
| Medication | 106.1 (103) | 202.2 (37) | 29.1–252.7 | 126.4–454.9 |
| Imaging | 13.9 (96) | 15.2 (12) | 5.1–39.2 | 7.0–25.3 |
| Radiotherapy | 25.3 (21) | N/A | 7.6–176.9 | N/A |
| Surgery | 126.4 (76) | 758.1 (25) | 48.0–252.7 | 758.1–909.7 |
| Pathology | 7.6(87) | 11.4 (35) | 4.3–12.6 | 11.4–45. |
| Chemotherapy | 151.6 (92) | 88.4 (51) | 75.8–252.7 | 29.9–202.2 |
| Inpatient Service | 75.8(76) | 278.0 (10) | 25.3–167.41 | 31.6–1,200.3 |
| All Services | 336.3 (145) | 926.1 (5) | 97.4–710.6 | 205.9–1,580.6 |

IQR: Interquartile range; USD: United States dollars; PCC: Tikur Anbessa Specialized Hospital-Radiation Centre; PHF: Private health care facilities; N/A: Not applicable.

of 758 USD (IQR 758 to 910) for surgery; 278 USD (IQR 32–1,200) for inpatient service including all medical treatment, a bed, and meals; and 202 USD (IQR 126 to 455) for medication (Table 2).

About 36% of patients obtained financial support from relatives/friends and 20% used combinations of financial support, savings, and loans for their treatment. However, 79% of patients reported having difficulty obtaining money.

## Willingness and ability to pay

151 patients treated at PCC had WATP median of 50 USD (IQR 19 to 134) and 53 patients treated at PHFs had WATP median 149 USD (IQR 66 to 383) for all health care services. Patients treated at PCC had WATP median 13 USD (IQR 2.5 to 37.27) for surgery, 13 USD (IQR 0.0 to 25.27) for chemotherapy and 13 USD (IQR 1.3 to 25.27) for inpatient service. The 13 USD was equivalent to 500 Birr and was higher than the median amounts of other health care services that were provided in PCC. Patients treated at PHFs were willing and able to pay a higher median amount of 253 USD (IQR 253 to 379) for surgery and 35 USD (IQR 23 to 76) for drugs (Table 3).

The costs incurred for treatment were analyzed by years since diagnosis. We found that the patients who were treated for up to two years paid the median amount of 505 USD, while patients who were treated for more than two years paid the median amount of 454 USD for the whole treatment period. The patients differed in their willingness and ability to pay depending on the duration of treatment (medians of 71 USD and 27 USD, respectively).

## Factors affecting WATP for BC treatment

The results of the multivariable regression are listed in Table 4 WATP increased with the total expenditure of patients, educational level and service quality. An increase of 100 USD in total expenditure corresponded to an increase by 43% in WATP (OR 1.43, 95% CI 1.09 to 1.89). Likewise, each increased educational or service quality level corresponded to an increase by 37% and 34% in WATP respectively (OR 1.37; 95% CI 1.02 to 1.85 and OR 1.34; 95% CI 1.04 to 1.72). An income increase of 100 USD corresponded to a 17% decrease of WATP (OR 0.83; 95% CI 0.70 to 0.99).

**Table 3. Comparison between PCC and PHFs concerning WATP for each component of the BC treatment (n = 204).**

| Service | Median (1) | | IQR (USD) | |
|---|---|---|---|---|
| | PCC | PHF | PCC | PHF |
| Consultation | 0.1 (151) | 5.1 (51) | 0.1–0.38 | 3.8–5.05 |
| Laboratory | 1.3 (133) | 12.6 (38) | 0.5–2.53 | 8.2–25.27 |
| Drugs | 7.6 (122) | 35.4 (31) | 2.5–25.27 | 22.7–75.81 |
| Imaging | 1.3 (123) | 9.1 (12) | 0.5–5.05 | 7.0–15.09 |
| Radiotherapy | 2.5 (94) | N/A | 1.1–12.63 | N/A |
| Surgery | 12.6 (118) | 252.7 (22) | 2.5–37.27 | 252.7–379.05 |
| Pathology | 1.3 (125) | 12.6 (19) | 0.8–2.53 | 11.4–21.33 |
| Chemotherapy | 12.6 (131) | 20.2 (33) | 0.0–25.27 | 12.6–50.54 |
| Inpatient service | 12.6 (97) | 25.3 (11) | 1.3–25.27 | 16.4–138.98 |
| All services | 49.8 (151) | 149.1 (53) | 19.1–134.0 | 65.7–382.8 |

IQR: Interquartile range; USD: United States dollars; PCC: Tikur Anbessa Specialized Hospital-Radiation Centre; PHF: Private health care facilities; Diff: difference between PHF and PCC medians; N/A: Not applicable.

## Discussion

The main aims of this study were to assess how much BC patients spent on treatment and patients' WATP for BC treatment in Addis Ababa, Ethiopia. The main finding of this study was that costs were considerably high at PHF and three times higher at PCC. The median amounts that patients eventually paid for all services both at PCC and PHFs were six times higher than the median amounts they stated being willing and able to pay both in PCC and PHF. This indicates that these patients were forced to pay a higher amount for treatment of a life-threatening disease regardless of their WATP. Our findings show that the actual cost is more than what patients are willing and able to pay and more than their demand for health care services. This apparent contradiction is likely due to poorer report concerning what patients remember having spent in the past.

**Table 4. Multivariable regression summary (dependent variable is WATP, in USD.**

| Variables | OR | 95% CI (lower) | 95% CI (upper) | p-value |
|---|---|---|---|---|
| Marital Status (Ref: Married, n = 86) | | | | |
| **Ex-married (n = 24)** | **0.20** | **0.11** | **0.34** | **< 0.001** |
| Single (n = 11) | 0.91 | 0.44 | 1.86 | 0.787 |
| Occupation (Ref: Housewife, n = 46) | | | | |
| Employed (n = 75) | 0.71 | 0.46 | 1.12 | 0.142 |
| Treatment duration (Ref: >2 years, n = 35) | | | | |
| ≤ 2 years (n = 86) | 0.63 | 0.39 | 1.02 | 0.062 |
| **Higher monthly income** (per 100 USD, n = 121) | **0.83** | **0.70** | **0.99** | **0.043** |
| Higher amount expended for BC treatment (per 100 USD, n = 121) | 1.00 | 0.99 | 1.00 | 0.792 |
| **Higher Total Expenditure** (per 100 USD, n = 121) | **1.43** | **1.09** | **1.89** | **0.010** |
| Higher Family Size (n = 121) | 0.90 | 0.81 | 1.01 | 0.070 |
| **Increasing Educational Level** (3 levels, n = 121) | **1.37** | **1.02** | **1.85** | **0.036** |
| **Increasing Service Quality** (3 levels, n = 121) | **1.34** | **1.04** | **1.72** | **0.026** |
| Higher Age (in years, n = 121) | 1.00 | 0.97 | 1.02 | 0.708 |

## High absolute costs of treatment

Total costs for treatment were far higher than WATP. Even if patients were economically poor and dependent on others, they had to pay high amounts for their BC treatment. Due to the high cost of treatment, 56% of interviewed patients as well as their families were exposed to catastrophic health expenditure and financial distress. Studies conducted in the Mandura District, Western Ethiopia and in Dessie Referral Hospital, Northeast Ethiopia indicated that 22.5% and 64.2% patients were exposed to catastrophic health expenditure due to direct medical costs respectively, though the participants in those studies were not cancer patients [42,43]. A study conducted in Nigeria also reported that the mean OOP expenditure for BC diagnosis and management was 2,049 USD [44]. According to a study that assessed the prices of health care in 15 African countries, including Ethiopia, patients were compelled to sell their fixed and precious assets and to borrow to cover their health care expenditures. Moreover, a high proportion of patients were not able to get health care services, because they could not afford the costs [45,46]. Other patients paid 2,325 USD in Addis Ababa for cancer treatment [47], but 600 USD in Uganda and 2100 USD in Nigeria [48]. Our finding is also compatible with the theoretical definition of catastrophic health care because the following factors are met: the cost of health care service is covered OOP, (2) the household CTP is low, and (3) there is no prepayment system for risk accumulation [11]. WHO also declared that a household is said to be exposed to catastrophic health expenditure if the OOP expenditure on health is greater than or equal to 40% of the household's CTP [49]. In other words, the health expenditure is considered catastrophic if it is above the financial means of the household and this will lead the household to poverty or prevent the household from getting out of poverty. As indicated in the present study, most of the patients were exposed to catastrophic health expenditure because they incurred costs beyond their CTP. This shows that even though Ethiopian patients are in a lower economic category than patients in other countries, they spent similar amounts of money for their BC treatment. This underlines the high economic burden of BC on Ethiopian patients.

Patients estimated their personal WATP but eventually retrieved funding from a much larger extended family. Most of the patients were housewives who were financially dependent on income from their husbands, children, and relatives. These payments for the patients' BC treatment created financial distress not just for the patient but for the extended family as well. A study conducted in Nigeria indicated that about 78% of study participants with BC were not able to get treatment because of financial barriers they faced [50] and research conducted in Iran indicated that due to the cost of cancer treatment, patients' and their families' living conditions were worsened [11].

## Comparison of private vs. public health care facilities

We observed that patients paid more at PHFs compared with PCC for similar treatment because PHFs are profit making institutions and provide treatment more quickly than PCC. For instance, a study conducted at PCC, Addis Ababa, Ethiopia and Yaoundé General Hospital, Cameroon revealed that patients with BC had long waiting times between their first consultation and surgery or other treatments [51,52]. In order to avoid treatment delays and to reduce the chance of the cancer progressing during the wait, patients preferred to pay more.

Various studies found that WATP can be influenced by factors such as age, education, income, dependency ratio/household size, attitude towards treatment, quality of health care services, locality rural/urban, and ability to pay [53]. In our study, we stated an association between average monthly income, education, service quality, and total expenditure and WATP of BC patients.

Increasing income was associated with decreased WATP. This finding was contrary to other studies that dealt with the WTP for BC treatment and other types of cancers because these studies focused on WTP only while our study focused on WATP [26,54,55]. Thus, we found that with increasing income, patients were more able but less willing to contribute to a higher amount of treatment costs. This result is consistent with studies conducted in Bangladesh, Sweden, and Ethiopia had found that the impact of income on WTP for medical care was negligible [56–59].

Educational level is a predictor of WATP. Studies showed that patients with higher educational level have better health literacy, resulting in them being better informed about the treatment and disease and more likely to be willing and able to pay for breast cancer treatment [29,60,61].

Perceived service quality was also associated with WATP. As patients perceived that the service quality is good, they were more willing and able to pay for their BC treatment. Especially economically better-off patients were willing and able to pay more for services that they believed were good. This is consistent with other studies that reported, patients who obtained good quality health care service were willing to pay more [53].

Total expenditure was also associated with WATP. This is because expenditure was directly related to income. This finding is consistent with the economic theory that a person's decision is a sequential process where the decision of whether to consume a particular commodity is followed by the choice of how much to consume. As health is a commodity and private good, this economic theory is also applicable to health care service, as we did in our study [62,63].

## Potential limitations of the study

Patients were frequently unaware of the stage of their disease and staging might have changed during treatment. Thus, we did not consider the exact stages of BC of the interviewed patients and only differentiated between patients with up to 2 years and more than 2 years of total duration of treatment. Responses about actual household income had a high degree of uncertainty as income may vary greatly over time and estimating an average was difficult for respondents. Additionally, the definition of the household by participant varied and the respondents included closer or more distant relatives. Since the IQR of WATP only varied five-fold, we assume that the group of relatives with financial contributions to WATP were probably perceived as relatively similar among respondents.

## Conclusion

The cost of BC treatment in Addis Ababa was beyond the patients' WATP and they were compelled to pay the required amount by asking for financial support from their relatives, including borrowing. Patients from public and private centres had similar disparities between WATP and actual costs indicating that eventually most patients and their families were exposed to catastrophic health expenditure. This indicated that the amount BC patients paid for their treatment was beyond their WATP, but they paid the amount indicated in this study because BC is life threating disease and patients and their families wanted to increase survival time. Thus, the contribution of this research is that decision makers in the health sector at all levels including at private health facilities should consider the WATP of BC patients when deciding on the fee for BC treatment.

## Recommendations

Therefore, seeing these catastrophic expenditures, costs need to be limited and communicated early to the patients. Policy makers in the health sector should include BC treatment in both

social and community-based health insurances. Also, health facilities should ensure that their health care services fees are reasonable. Creative ways to co-finance from additional sources are needed to meet the objective of the WHO Global Breast Cancer Initiative assuring that 80% of the patients complete multimodal treatment.

## Acknowledgments

We thank all collaborators including the employees of the public and private health facilities where data were collected and BC patients who were willing to be interviewed. Special thanks go to the School of Public Health, College of Health Science Addis Ababa University and Martin Luther University Halle-Wittenberg, Germany, for their major contribution to this study by providing the necessary technical and logistic support during the study as required. We acknowledge the proofreading assistance of Dawn M. Bielawski, PhD.

## Author Contributions

**Conceptualization:** Tamiru Demeke, Damen Hailemariam, Edom Seife, Adamu Addissie, Rafael Mikolajczyk, Eva Johanna Kantelhardt.

**Data curation:** Tamiru Demeke.

**Formal analysis:** Tamiru Demeke, Damen Hailemariam, Pablo Santos, Adamu Addissie, Rafael Mikolajczyk, Eva Johanna Kantelhardt, Susanne Unverzagt.

**Investigation:** Tamiru Demeke, Adamu Addissie, Rafael Mikolajczyk, Eva Johanna Kantelhardt, Susanne Unverzagt.

**Methodology:** Tamiru Demeke, Damen Hailemariam, Pablo Santos, Adamu Addissie, Rafael Mikolajczyk, Eva Johanna Kantelhardt, Susanne Unverzagt.

**Project administration:** Tamiru Demeke, Eva Johanna Kantelhardt.

**Resources:** Eva Johanna Kantelhardt.

**Software:** Tamiru Demeke, Pablo Santos.

**Supervision:** Tamiru Demeke, Damen Hailemariam, Pablo Santos, Adamu Addissie, Rafael Mikolajczyk, Eva Johanna Kantelhardt, Susanne Unverzagt.

**Validation:** Tamiru Demeke, Pablo Santos, Edom Seife, Rafael Mikolajczyk, Eva Johanna Kantelhardt, Susanne Unverzagt.

**Visualization:** Tamiru Demeke, Pablo Santos, Eva Johanna Kantelhardt, Susanne Unverzagt.

**Writing – original draft:** Tamiru Demeke.

**Writing – review & editing:** Tamiru Demeke, Eric Sven Kroeber, Rafael Mikolajczyk, Birgit Silbersack, Eva Johanna Kantelhardt, Susanne Unverzagt.

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
