## [Decision Letter · Decision Letter 0]

12 Feb 2024

PONE-D-23-33162Willingness and ability to pay for breast cancer treatment among patients from Addis Ababa, Ethiopia: a cross-sectional studyPLOS ONE

Dear Dr. Unverzagt,

Thank you for submitting your manuscript to PLOS ONE. After careful consideration, we feel that it has merit but does not fully meet PLOS ONE’s publication criteria as it currently stands. Therefore, we invite you to submit a revised version of the manuscript that addresses the points raised during the review process.

Please, consider all comments of all reviewers  

We look forward to receiving your revised manuscript.

Kind regards,

Ahmed Mancy Mosa, Ph.D.

Academic Editor

PLOS ONE

[We thank all collaborators including the employees of the public and private health facilities where data were collected, BC patients who were willing to be interviewed and DAAD PAGEL project No. 57513614, Else-Kroener-Foundation, German Ministry of Research and Education that provided support in various ways for successful accomplishment of this study. Special thanks go to the School of Public Health, College of Health Science Addis Ababa University and Martin-Luther-University, Halle-Wittenberg, Germany for their major contribution to this study by providing the necessary technical and logistic support during the study as required. We acknowledge the proofreading assistance of Dawn M. Bielawski, PhD.]

 [This work was supported by DAAD PAGEL project No. 57513614. It was also supported by Else-Kroener-Foundation through Martin-Luther-University, Halle-Wittenberg, Germany, Grant No. 2018_HA31SP. The work was also supported through the German Ministry of Research and Education, Grant 01KA2220B. The funders had no role in study design, data collection and analysis, decision to publish, or preparation of the manuscript]

3. PLOS requires an ORCID iD for the corresponding author in Editorial Manager on papers submitted after December 6th, 2016. Please ensure that you have an ORCID iD and that it is validated in Editorial Manager. To do this, go to ‘Update my Information’ (in the upper left-hand corner of the main menu), and click on the Fetch/Validate link next to the ORCID field. This will take you to the ORCID site and allow you to create a new iD or authenticate a pre-existing iD in Editorial Manager. Please see the following video for instructions on linking an ORCID iD to your Editorial Manager account: " ext-link-type="uri" xlink:type="simple">https://www.youtube.com/watch?v=_xcclfuvtxQ".

Reviewers' comments:

Reviewer's Responses to Questions

**Comments to the Author**

1. Is the manuscript technically sound, and do the data support the conclusions?

Reviewer #1: Yes

Reviewer #2: Yes

Reviewer #3: Yes

2. Has the statistical analysis been performed appropriately and rigorously? 

Reviewer #1: Yes

Reviewer #2: Yes

Reviewer #3: Yes

3. Have the authors made all data underlying the findings in their manuscript fully available?

Reviewer #1: Yes

Reviewer #2: Yes

Reviewer #3: Yes

4. Is the manuscript presented in an intelligible fashion and written in standard English?

Reviewer #1: Yes

Reviewer #2: Yes

Reviewer #3: Yes

5. Review Comments to the Author

Reviewer #1: Manuscript “Willingness and ability to pay for breast cancer treatment among patients from Addis Ababa, Ethiopia: a cross-sectional study” has described the reality of Breast Cancer treatment in Ethiopia. The manuscript is well written and data provided support the author’s conclusion.

Reviewer #2: Comments to the Author

Congratulations on the submitted manuscript. The topic is timely and will be of interest to the readers of the journal. However, a few changes are suggested to improve the clarity of this manuscript. I have several recommendations and questions about the manuscript.

1.Abstract

Methods: We collected data from 204 randomly selected BC patients who received

treatment at one of four different health facilities (one public and three private) between

April 2020 and January 2021.

Recommend replacing the word “we” with “researchers”

What are the inclusion and exclusion criteria?

2. Introduction

Very good explanation

3. Methodology

Please state the sample design used in this study.

4. Recommendation

Recommend adding the subheading for recommendations for this study.

5.Conclusion

How do the findings add to the body of scientific knowledge on the particular issue?

Should highlight the implication of the results to health policy.

6. References

Most of the citations are less than 5 years. 39/61

Recommendation: 5 years above is more updated.

Please add doi in the article’s references.

Reviewer #3: Overall, the article is well-structured with distinct sections for abstract, introduction, methods, results, and conclusions. This makes it easy to follow.

1. Line 59; consider rephrasing "According to the Addis Ababa Population Based Cancer Registry database" to "According to data from the Addis Ababa Population Based Cancer Registry."

2. Line 63, instead of "Many BC patients come to Tikur Anbessa Specialized Hospital-Radiation Centre," you could write "Many breast cancer (BC) patients seek treatment at the Tikur Anbessa Specialized Hospital-Radiation Centre."

3. Line 74; consider rephrasing "This holds true for BC patients" to "This is also applicable to BC patients."

4. Line 89, it would be clearer to replace "To determine whether the treatment is affordable or not" with "To determine the affordability of treatment."

5. Line 90, "Willingness to pay (WTP) is defined as a person's willingness to pay for the good or services he/she wants to buy while ability to pay (ATP) is defined as the person's ability to pay for the chosen good or services" could be simplified to "Willingness to pay (WTP) refers to a person's willingness to pay for desired goods or services, while ability to pay (ATP) refers to their financial capacity to do so.

6. Line 102; consider rephrasing "Examining WATP is central to a meaningful analysis of demand for health care services" to "Analyzing WATP is crucial for understanding the demand for health care services."

7. While you've mentioned that the sample size was distributed among the four health facilities based on the number of BC patients they treated before data collection, it would be beneficial to provide more details on how this distribution was carried out. Was it a simple proportional allocation, or were there other considerations involved?

8. It's good that you accounted for a non-response rate of 15%. However, providing a brief justification or citing literature for choosing this specific rate would enhance the transparency of your methodology.

9. The assumed standard deviation of direct medical costs ($355 USD per BC treatment) seems crucial for sample size calculation. Providing a justification or reference for this assumption would strengthen the validity of your calculations.

10. It's mentioned that imputations were not performed for missing data. While this is acceptable, it would be useful to mention the extent of missing data and any sensitivity analyses conducted to assess the impact of missing data on the results.

11. Line 114 "For this study, we included all BC patients who were permanently residing in Addis Ababa and who were receiving BC treatment in PCC and the three privately owned health facilities at the time of data collection." - To improve clarity, consider rephrasing: "For this study, we included all breast cancer (BC) patients permanently residing in Addis Ababa and receiving treatment at PCC and the three privately owned health facilities at the time of data collection."

12. Line 149 "Informed patient consent was obtained orally at the beginning of each interview and documented on the questionnaire by the data collectors." - Consider revising for clarity: "Informed consent was obtained orally from each patient at the beginning of the interview and documented on the questionnaire by the data collectors."

13. In Table 1 Explanation of Income Categories: Provide a brief explanation of the income categories (e.g., low-income, middle-income, high-income) to give context to readers who may not be familiar with the currency or income distribution in the study setting

14. In Table 4, the sample sizes for some categories (e.g., unemployed) are very small, which could affect the reliability of estimates.

15. It might be beneficial to compare the costs reported in this study with national or regional averages for breast cancer treatment. This comparison could help contextualize whether the observed costs are consistent with broader trends in healthcare pricing.

16. Line 271; there is number 3 between ( ); it stands for what?

17. Line 245; the numbers order is not corrected, Please fix it?

6. PLOS authors have the option to publish the peer review history of their article (what does this mean?). If published, this will include your full peer review and any attached files.

Reviewer #1: No

Reviewer #2: **Yes: **DR RUSNANI AB LATIF

Reviewer #3: No

---

## [Author Response · Author response to Decision Letter 0]

22 Feb 2024

Reviewer #2: Comments to the Author

1.Abstract

Methods: We collected data from 204 randomly selected BC patients who received

treatment at one of four different health facilities (one public and three private) between

April 2020 and January 2021.

Comment #1

Recommend replacing the word “we” with “researchers”

Answer of the authors: Action taken as recommended. 

Comment #2

What are the inclusion and exclusion criteria?

Answer of the authors:

We tried to include the following statement in the abstract, but it will make the abstract to have words that exceed the limit i.e. the number of words of the abstract will be 308 while the word limit was 300 only. For this reason, we cannot add it but this statement is explained in the main body of the manuscript in method section. 

The inclusion criteria were that being BC patient, residing in Addis Ababa and receiving treatment at the time of data collection as described in the objectives and the first sentence of the methods.

2. Introduction

Comment #3

Very good explanation

Answer of the authors: Thank you

3. Methodology

Comment #4

Please state the sample design used in this study.

Answer of the authors:

We added the following statement as suggested in the materials and method section and write now: “The sample design was simple random sampling that is BC patients who came to the aforementioned health facilities will be randomly selected among those BC patients who came to the health facilities at the time of data collection.”

4. Recommendation

Comment #5

Recommend adding the subheading for recommendations for this study.

Answer of the authors:

The word “Recommendations” is added as subheading 

5.Conclusion

Comment #6

How do the findings add to the body of scientific knowledge on the particular issue?

Should highlight the implication of the results to health policy.

Answer of the authors:

We added the following paragraph in the conclusion:

“Thus, the contribution of this research is that decision makers in the health sector at all levels including at private health facilities should consider the WATP of BC patients when deciding on the fee for BC treatment.”

6. References

Comment #7

Most of the citations are less than 5 years. 39/61

Recommendation: 5 years above is more updated.

Answer of the authors:

Yes, we agree with your comment. However, since the research topic i.e. WATP for breast cancer treatment is a rare topic on which very few researches have been conducted especially in the recent times. For this reason, we tried to find papers on this and related topics. Unfortunately, we were able to get only 27 (43.5%) papers that were published during the last five years. For clarity purpose, we tried to categorized the papers by five years gap and found that 27 (43.5%) of the manuscripts were published between 2018 and 2023 while 8 (12.9%) and 10 (16.1%) manuscripts were published between 2014 and 2017, 2009 and 2013 respectively. The remaining papers i.e. 17 (27.4%) were published between 1986 and 2008. .

Comment #7

Please add doi in the article’s references.

Answer of the authors:

We added the ‘DOI’ of almost all manuscripts except very few manuscripts that do not have DOI.

Reviewer #3

Comments: 

Overall, the abstract is well-structured with distinct sections for abstract, introduction, methods, results, and conclusions. This makes it easy to follow. 

1. Line 59; consider rephrasing "According to the Addis Ababa Population Based Cancer Registry database" to "According to data from the Addis Ababa Population Based Cancer Registry."

Answer of the authors:

Thanks for this advice. The recommended statement replaced the previous statement.

2. Line 63, instead of "Many BC patients come to Tikur Anbessa Specialized Hospital-Radiation Centre," you could write "Many breast cancer (BC) patients seek treatment at the Tikur Anbessa Specialized Hospital-Radiation Centre."

Answer of the authors:

Thanks for this advice. The recommended statement replaced the previous statement.

3. Line 74; consider rephrasing "This holds true for BC patients" to "This is also applicable to BC patients."

Answer of the authors:

Thanks for this advice. The recommended statement replaced the previous statement.

4. Line 89, it would be clearer to replace "To determine whether the treatment is affordable or not" with "To determine the affordability of treatment."

Answer of the authors:

Thanks for this advice. The recommended statement replaced the previous statement.

5. Line 90, "Willingness to pay (WTP) is defined as a person's willingness to pay for the good or services he/she wants to buy while ability to pay (ATP) is defined as the person's ability to pay for the chosen good or services" could be simplified to "Willingness to pay (WTP) refers to a person's willingness to pay for desired goods or services, while ability to pay (ATP) refers to their financial capacity to do so.

Answer of the authors:

Thanks for this advice. The recommended statement replaced the previous statement.

6. Line 102; consider rephrasing "Examining WATP is central to a meaningful analysis of demand for health care services" to "Analyzing WATP is crucial for understanding the demand for health care services."

Answer of the authors:

Thanks for this advice. The recommended statement replaced the previous statement.

7. While you've mentioned that the sample size was distributed among the four health facilities based on the number of BC patients they treated before data collection, it would be beneficial to provide more details on how this distribution was carried out. Was it a simple proportional allocation, or were there other considerations involved?

Answer of the authors:

The recommended statement is added as shown below: “This sample size was distributed among the four health facilities in proportion to the number of BC patients they treated before data collection time using simple proportional allocation method.”

8. It's good that you accounted for a non-response rate of 15%. However, providing a brief justification or citing literature for choosing this specific rate would enhance the transparency of your methodology.

Answer of the authors:

Non-response rate of 15% was chosen based on the experiences that we had from our previous researches. 

9. The assumed standard deviation of direct medical costs ($355 USD per BC treatment) seems crucial for sample size calculation. Providing a justification or reference for this assumption would strengthen the validity of your calculations.

Answer of the authors:

The amount USD 355 per BC treatment was obtained by converting ETB 5140.10 using the exchange rate USD 1= ETB 14.50. ETB 5140 was a standard deviation of direct medical cost in Tikur Anbessa Specialized Hospital. This figure was taken from a research (HailuMariam 2013. Patient side cost and its predictors for cervical cancer in Ethiopa: a cross sectional hospital based study) which was conducted in the aforementioned hospital. Due to the lack of regional studies to estimate the costs of breast cancer therapy, a study on the costs of cervical cancer therapy was used and the costs of both therapies were assumed to be comparable.

10. It's mentioned that imputations were not performed for missing data. While this is acceptable, it would be useful to mention the extent of missing data and any sensitivity analyses conducted to assess the impact of missing data on the results.

Answer of the authors:

We did consider imputation as a method to increase sample size. We decided against it because a correlation analysis (Pearson linear correlation) yielded only low levels of correlation (all rho 0.5), which is insufficient for imputation with high confidence. We understand this to be less “harmful” to the regression analysis (basically leading to larger confidence intervals around the estimates, leaning towards a type-II error) than the imputation of, in the more extreme of cases, 58 data points concerning “Average Monthly Income”. The latter could potentially lead to type-I errors. We have now added one sentence to the end of the Statistics section to clarify this, as well as the amount of data missing in each variable. 

11. Line 114 "For this study, we included all BC patients who were permanently residing in Addis Ababa and who were receiving BC treatment in PCC and the three privately owned health facilities at the time of data collection." - To improve clarity, consider rephrasing: "For this study, we included all breast cancer (BC) patients permanently residing in Addis Ababa and receiving treatment at PCC and the three privately owned health facilities at the time of data collection."

Answer of the authors:

Thanks for this advice. The recommended statement replaced the previous statement.

12. Line 149 "Informed patient consent was obtained orally at the beginning of each interview and documented on the questionnaire by the data collectors." - Consider revising for clarity: "Informed consent was obtained orally from each patient at the beginning of the interview and documented on the questionnaire by the data collectors."

Answer of the authors:

Thanks for this advice. The recommended statement replaced the previous statement.

13. In Table 1 Explanation of Income Categories: Provide a brief explanation of the income categories (e.g., low-income, middle-income, high-income) to give context to readers who may not be familiar with the currency or income distribution in the study setting 

Answer of the authors:

Explanation was added as per the reviewer’s comment as shown below:

“Participants in the study were grouped according to their monthly income based on the World Bank's Poverty Assessment of Ethiopia and Ethiopia's low income tax rate. Accordingly, an income of less than USD 1.25 per day was considered the poverty line (41). Monthly income greater than USD 38 but less than or equal to USD 75 is considered lower-middle income, monthly income greater than USD 75 but less than or equal to USD 150 is considered middle income, and monthly income greater than USD 150 is considered upper income.”

14. In Table 4, the sample sizes for some categories (e.g., unemployed) are very small, which could affect the reliability of estimates.

Answer of the authors:

We agree that considering a category such as “unemployed” with 2 data points has no positive contribution to the analysis. We have now grouped “unemployed” with “housewifes”. In this sense, the variable “occupation” has now a binary character of a woman being employed or not. We have re-run the multivariable regression and updated all results in table 4 and text. The regression with the new “occupation” variable leads to minimal changes in nearly all other estimates, p-values and confidence intervals, but does not have any influence on the interpretation of the results.

15. It might be beneficial to compare the costs reported in this study with national or regional averages for breast cancer treatment. This comparison could help contextualize whether the observed costs are consistent with broader trends in healthcare pricing.

Answer of the authors:

There are only few national or regional manuscripts on costs, but we cited them, e.g. in the subchapter “High absolute costs of treatment”:

“Studies conducted in the Mandura District, Western Ethiopia and in Dessie Referral Hospital, Northeast Ethiopia indicated that 22.5% and 64.2% patients were exposed to catastrophic health expenditure due to direct medical costs respectively, though the participants in those studies were not cancer patients (42, 43). A study conducted in Nigeria also reported that the mean OOP expenditure for BC diagnosis and management was 2,049 USD (44). According to a study that assessed the prices of health care in 15 African countries, including Ethiopia, patients were compelled to sell their fixed and precious assets and to borrow to cover their health care expenditures. Moreover, a high proportion of patients were not able to get health care services, because they could not afford the costs (45, 46). Other patients paid 2,325 USD in Addis Ababa for cancer treatment (47), but 600 USD in Uganda and 2100 USD in Nigeria (48).”

And 

“A study conducted in Nigeria indicated that about 78% of study participants with BC were not able to get treatment because of financial barriers they faced (50) and research conducted in Iran indicated that due to the cost of cancer treatment, patients’ and their families’ living conditions were worsened (11).”

As well as under subchapter "Comparison of private vs public health care facilities"

“For instance, a study conducted at PCC, Addis Ababa, Ethiopia and Yaoundé General Hospital, Cameroon revealed that patients with BC had long waiting times between their first consultation and surgery or other treatments (51, 52).”

“Thus, we found that with increasing income, patients were more able but less willing to contribute to a higher amount of treatment costs. This result is consistent with studies conducted in Bangladesh, Sweden, and Ethiopia had found that the impact of income on WTP for medical care was negligible (56-59).”

16. Line 271; there is number 3 between ( ); it stands for what? 

Answer of the authors:

Thank you, this was wrong. Even we do not know where it comes from. Thus, we deleted.

17. Line 245; the numbers order is not corrected, Please fix it? 

Answer of the authors:

Correction is made as per the instruction.

Thank you, the end

---

## [Decision Letter · Decision Letter 1]

4 Mar 2024

Willingness and ability to pay for breast cancer treatment among patients from Addis Ababa, Ethiopia: a cross-sectional study

PONE-D-23-33162R1

Dear Dr. Unverzagt,

We’re pleased to inform you that your manuscript has been judged scientifically suitable for publication and will be formally accepted for publication once it meets all outstanding technical requirements.

Kind regards,

Ahmed Mancy Mosa, Ph.D.

Academic Editor

PLOS ONE

Additional Editor Comments (optional):

Reviewers' comments:

Reviewer's Responses to Questions

**Comments to the Author**

1. If the authors have adequately addressed your comments raised in a previous round of review and you feel that this manuscript is now acceptable for publication, you may indicate that here to bypass the “Comments to the Author” section, enter your conflict of interest statement in the “Confidential to Editor” section, and submit your "Accept" recommendation.

Reviewer #2: All comments have been addressed

Reviewer #3: All comments have been addressed

2. Is the manuscript technically sound, and do the data support the conclusions?

Reviewer #2: Yes

Reviewer #3: Yes

3. Has the statistical analysis been performed appropriately and rigorously? 

Reviewer #2: Yes

Reviewer #3: Yes

4. Have the authors made all data underlying the findings in their manuscript fully available?

Reviewer #2: (No Response)

Reviewer #3: Yes

5. Is the manuscript presented in an intelligible fashion and written in standard English?

Reviewer #2: Yes

Reviewer #3: Yes

6. Review Comments to the Author

Reviewer #2: Dear Authors

Congratulations on your manuscript, I have checked the reviewer’s comment, and all the amendments were implemented in response to the reviewers' recommendations and commendations.

Reviewer #3: the authors have addressed all comments provided previously in the first round of revision. I am in favor of accepting the manuscript for publishing.

7. PLOS authors have the option to publish the peer review history of their article (what does this mean?). If published, this will include your full peer review and any attached files.

Reviewer #2: **Yes: **DR RUSNANI AB LATIF

Reviewer #3: No

---

## [Editor Report · Acceptance letter]

18 Mar 2024

PONE-D-23-33162R1 

PLOS ONE

Dear Dr. Unverzagt, 

I'm pleased to inform you that your manuscript has been deemed suitable for publication in PLOS ONE. Congratulations! Your manuscript is now being handed over to our production team.

Kind regards, 

on behalf of

Dr. Ahmed Mancy Mosa 

Academic Editor

PLOS ONE